# Spotlight on Exosomal Non-Coding RNAs in Breast Cancer: An In Silico Analysis to Identify Potential lncRNA/circRNA-miRNA-Target Axis

**DOI:** 10.3390/ijms23158351

**Published:** 2022-07-28

**Authors:** Ohanes Ashekyan, Samira Abdallah, Ayman Al Shoukari, Ghada Chamandi, Hayat Choubassy, Abdul Rahman S. Itani, Nisreen Alwan, Rihab Nasr

**Affiliations:** 1Department of Biochemistry and Molecular Genetics, Faculty of Medicine, American University of Beirut, Beirut 11-0236, Lebanon; oha09@mail.aub.edu; 2Department of Anatomy, Cell Biology and Physiological Sciences, Faculty of Medicine, American University of Beirut, Beirut 11-0236, Lebanon; sma145@mail.aub.edu (S.A.); gc21@aub.edu.lb (G.C.); hc59@aub.edu.lb (H.C.); 3Department of Experimental Pathology, Immunology, and Microbiology, Faculty of Medicine, American University of Beirut, Beirut 11-0236, Lebanon; aza23@mail.aub.edu; 4INSERM U976, HIPI, Pathophysiology of Breast Cancer Team, Université de Paris, 75010 Paris, France; 5Faculty of Sciences, Lebanese University, Beirut 11-0236, Lebanon; 6Faculty of Biosciences, Heidelberg University, 69120 Heidelberg, Germany; abed.s.itani@gmail.com; 7Heidelberg Institute for Stem Cell Technology and Experimental Medicine (HI-STEM gGmbH), 69120 Heidelberg, Germany; 8Division of Inflammatory Stress in Stem Cells, Deutsches Krebsforschungszentrum (DKFZ) and DKFZ-ZMBH Alliance, 69120 Heidelberg, Germany; 9College of Health Sciences, Abu Dhabi University, Abu Dhabi 59911, United Arab Emirates

**Keywords:** breast cancer, exosomes, non-coding RNA, microRNAs, long non-coding RNA, circular RNAs, biomarkers

## Abstract

Breast cancer (BC) has recently become the most common cancer type worldwide, with metastatic disease being the main reason for disease mortality. This has brought about strategies for early detection, especially the utilization of minimally invasive biomarkers found in various bodily fluids. Exosomes have been proposed as novel extracellular vesicles, readily detectable in bodily fluids, secreted from BC-cells or BC-tumor microenvironment cells, and capable of conferring cellular signals over long distances via various cargo molecules. This cargo is composed of different biomolecules, among which are the novel non-coding genome products, such as microRNAs (miRNAs), long non-coding RNAs (lncRNAs), and the recently discovered circular RNA (circRNA), all of which were found to be implicated in BC pathology. In this review, the diverse roles of the ncRNA cargo of BC-derived exosomes will be discussed, shedding light on their primarily oncogenic and additionally tumor suppressor roles at different levels of BC tumor progression, and drug sensitivity/resistance, along with presenting their diagnostic, prognostic, and predictive biomarker potential. Finally, benefiting from the miRNA sponging mechanism of action of lncRNAs and circRNAs, we established an experimentally validated breast cancer exosomal non-coding RNAs-regulated target gene axis from already published exosomal ncRNAs in BC. The resulting genes, pathways, gene ontology (GO) terms, and Kyoto Encyclopedia of Genes and Genomes (KEGG) analysis could be a starting point to better understand BC and may pave the way for the development of novel diagnostic and prognostic biomarkers and therapeutics.

## 1. Introduction

According to the International Agency for Research on Cancer (IARC), in 2020, breast cancer (BC) became the most prevalent cancer type worldwide, with more than 2.26 million new cases and nearly 685,000 deaths, most of which being a cause of metastatic disease. Moreover, BC was the most common cause of cancer-related deaths in women and the fifth most common cause of cancer death overall [1]. This increase, as well as the initiation and progression of the disease, could be attributed to various genetic, reproductive, or demographic risk factors. Early detection techniques available today include mammography, ultrasound, magnetic resonance imaging, positron emission tomography, and breast biopsies [2]. However, these techniques have limitations in terms of their accuracy, cost, duration, and ease of use that limit their efficacy. As such, there is an urge to unveil novel, robust, inexpensive, and minimally invasive biomarkers to ascertain BC at an early stage and circumvent poor disease prognosis.

Discovering BC diagnostic and prognostic biomarker strategies have been increasingly utilized as a means of early detection and avoidance of bad disease prognosis. Accordingly, taking advantage of exosomes, which are cell-derived extracellular vesicles of endosomal origin ranging in size from 30–100 nm [3] found in body fluids, i.e., saliva, urine, serum, amniotic fluid, cerebrospinal fluid, bile, etc. [4,5,6,7,8,9], has become an insightful approach, especially with the ease of their detection in liquid biopsies. Currently, exosomes are attracting tremendous attention for their important roles in cellular communication by enabling the transfer of various types of cargo, such as proteins, mRNA, DNAs, and non-coding RNAs, wrapped in a lipid bilayer, to recipient cells [3,4]. Among the different types of cargo, non-coding RNAs are the ones attracting particular attention due to their promising clinical applications, such as potential biomarkers in cancer diagnosis and prognosis. 

The traditional notion of gene regulation is centered around the central dogma, which proposes that DNA is transcribed into mRNAs, which are, in turn, translated into proteins. However, recently, discoveries through several high-throughput genomic platforms suggest that non-coding portions of the genome are also responsible for the tight regulation of gene expression. It was demonstrated that just as little as <1% of the human genome encodes proteins and that a large proportion of the genome is actively transcribed into non-coding RNAs (ncRNAs) [10]. The well-studied ncRNAs are micro-RNAs (miRNAs), which are small ncRNAs of ~20–23 nucleotides in length, and long non-coding RNAs (lncRNAs), which are defined as transcripts longer than 200 nucleotides. Whereas miRNAs regulate gene expression primarily through mRNA degradation or silencing, lncRNAs act via several mechanisms at both the transcriptional and translational levels [11]. In addition, another recently understood significant class of ncRNAs are the circular RNAs (circRNAs), which are closed-end long non-coding RNAs, where their 5′ and 3′ ends are covalently attached via a process called “back-splicing” [12]. CircRNAs are presented to be more stable and resistant to degradation by exonucleases than most linear RNAs since they do not harbor a 5′ or 3′ end, and are believed to exert analogous roles as lncRNAs [13]. The abovementioned ncRNAs are essential for the regulation of various biological processes and were shown to be dysregulated in several types of cancers, including BC [14]. In addition, they could have a possible interaction, bringing about a higher level of regulation and a novel mechanism of action conceptualized as miRNA sponging, where lncRNAs and circRNAs bind and inhibit the functionality of miRNAs [11].

Exosomes were first reported to carry RNAs by Valadi et al. [15], where they were demonstrated to be carrying mRNAs and miRNAs. Later on, evidence of other types of ncRNA cargo in exosomes, such as lncRNAs [16] and circular RNAs [17], was also reported. Exosomes readily shelter packaged cargo molecules from enzyme-mediated destruction. They increase the circulating half-life and thus enhance the downstream effects mediated by the transfer of their various types of cargo molecules—specifically, ncRNAs-into other distant cellular entities [18]. LncRNAs and circRNAs that are capable of interacting with miRNAs are termed competitors of endogenous RNAs (ceRNAs). CeRNAs harbor microRNA response element (MRE) sequences, which permit their binding to miRNAs. CeRNAs sponge miRNAs, preventing them from exerting their normal function of inhibiting protein-coding mRNA translation through MRE complementarity (Figure 1). CeRNAs play various roles in different phases of cancer, where they may act as oncogenes or tumor suppressor genes depending on the function of the protein-coding gene downstream of the target miRNA [19,20,21].

Vast literature evidence is present regarding the existence of ncRNA cargo in exosomes of the bodily fluids derived from breast cancer cells. Moreover, there is evidence of ncRNA cargo in exosomes derived from breast cancer tumor microenvironment (TME) cells, such as cancer-associated fibroblasts (CAFs) [22,23,24,25,26,27,28,29]. This brings about the notion that not only can breast cancer cell-derived exosomes alter TME cells, but also that TME-derived exosomes may alter breast cancer cells, and this is primarily achieved via the ncRNA cargo of exosomes. Thus, exosomes from the collective breast tumor cells and the microenvironment should be taken into consideration as mediators of carcinogenesis.

In this review, the various roles of the three types of ncRNA cargo of breast cancer and TME-derived exosomes will be thoroughly investigated, stressing their overall oncogenic or tumor suppressor roles in various levels of tumorigenesis and drug sensitivity/resistance. In addition, the diagnostic, prognostic, and predictive potential of these three ncRNA types will be presented. Lastly, an in silico approach will be adopted, taking advantage of the miRNA sponging mechanism of action of lncRNAs and circRNAs to establish an experimentally validated breast cancer exosomal non-coding RNAs-regulated target gene axis. This approach may be utilized to better describe the diverse effects of exosomal non-coding RNAs in breast cancer and to manipulate this axis to develop novel therapeutics.

## 2. Oncogenic and Tumor Suppressor Roles of Exosomal Non-Coding RNAs in BC

Exosomes were shown to carry and transport various types of ncRNAs, including miRNAs, lncRNAs, and circRNAs. In addition, BC cell and TME-derived exosomes were demonstrated to carry and transport ncRNAs. Thus, taking into consideration the diverse roles that these ncRNAs can play at different levels, such as at the transcriptional, translational, and epigenetic levels, they were presented to harbor oncogenic or tumor suppressor potential, affecting BC progression and drug resistance.

### 2.1. Roles of Exosomal ncRNAs in BC Cell Growth and Proliferation

Exosomal miRNA has been shown as an important player in promoting or inhibiting oncogenesis. A study has demonstrated the tumor-suppressive roles of miR-145 by modulating ROCK1, MMP9, ERBB2, and TP53 gene expression [30]. It was also reported by Yan et al. that miR-105 elevates MYC protein levels in cancer-associated fibroblasts. The increased expression of miR-105 and MYC enhanced glycolysis and increased the nutrient use. The knockdown of MYC expression abolished miR-105- and MYC-induced nutrient metabolism. This study demonstrated a MYC–miR-105–MXI1–MYC loop that leads to miR-105-reprogramming in CAFs nourishing cancer cells with energy-rich metabolites and contributing to sustained tumor growth [31]. Moreover, Li et al. have shown a potential for triple negative breast-cancer-derived exosomal miR-1246 in promoting tumor growth in normal human normal epithelial (HMLE) cells by targeting CCNG2 [32]. In another study, Jung et al. validated that exosomal miR-210 could be transferred from hypoxic breast-cancer-derived exosomes and promote angiogenesis and tumor growth in recipient normal cells in the tumor microenvironment by acting on vascular remodeling-related genes Ephrin A3 and PTP1B [33].

Exosomal lncRNAs were also shown to act as oncogenes or tumor suppressor genes via their most robust miRNA sponging action and other mechanisms. The lncRNA X-inactive-specific transcript (XIST) was shown to harbor a tumor suppressor potential in breast cancer through its exosomal miR-503 sponging ability, where knockdown strategies of XIST revealed the promotion of malignancy and stemness through the relief of exosomal miR-503 sponging [34]. On the other hand, lncRNA SNHG3 was demonstrated to exert an oncogenic role in breast cancer (BC) by facilitating a metabolic reprogramming event. Exosomal SNHG3 was shown to act via sponging miR-330-5p, relieving its inhibitory effect on its target gene pyruvate kinase M1 (PKM). Collectively, CAFs-secreted exosomal SNHG3 resulted in a decrease in mitochondrial oxidative phosphorylation and an increase in glycolysis carboxylation, leading to an increased breast cancer cell growth. In vitro and in vivo CAFs-secreted exosomal SNHG3 knockdown strategies demonstrated a reversal of the metabolic reprogramming, leading to a decreased breast cancer cell glycolysis and growth [25].

In addition, the oncogenic effect of exosomal circRNAs has been verified by multiple studies. Zhang et al. showed that exosomal circFOXK2 facilitates oncogenesis in breast cancer via interacting with IGF2BP3 and miR-370. The exosomal circFOXK2 level was significantly increased in breast cancer cells with a high metastatic ability, and its overexpression promoted the migration and invasion of BC cells [35]. In another study, exosomal circHIF1A significantly promoted triple negative breast cancer growth. An exosomal circHIF1A/NFIB/FUS feedback loop was validated by Chen et al., as exosomal circHIF1A modulated the expression and translocation of NFIB through post-transcriptional and post-translational modification. FUS was found to be able to regulate the biogenesis of exosomal circHIF1A by interacting with the flanking intron, and FUS was transcriptionally regulated by NFIB [36].

### 2.2. Roles of Exosomal ncRNAs in BC Metastasis

The intercellular communicator role of exosomes has made them a promising research entity for their potential metastasis-promoting and metastasis-inhibiting abilities while affecting the cells of the tumor microenvironment. Markedly, the non-coding RNA cargo of exosomes is believed to enable the promotion or inhibition of metastasis via epigenetic regulation [37] Literature evidence revealed that exosomal miRNAs could be transferred from tumor-derived exosomes into normal cells, promoting metastasis. MiR-500a-5p transferred from CAFs-derived exosomes was shown to enhance breast cancer metastasis by binding to ubiquitin-specific peptidase 28 (USP28) [22]. Tumor-derived exosomal miR-7641 was demonstrated to promote metastasis via intercellular communication [38]. Transferred miR-1246, expressed tremendously in metastatic breast cancer MDA-MB-231, was exhibited to promote invasion in normal HMLE cells by targeting CCNG2 [32]. Hypoxic breast cancer-derived exosomal miR-210 was proved to escalate into adjacent cells in the tumor microenvironment, promoting angiogenesis and metastasis in recipient cells by targeting Ephrin A3 and PTP1B, which are vascular remodeling-related genes [33]. MiR-1910-3p, transferred from breast cancer cell-derived exosomes to normal mammary epithelial cells, was validated to enhance metastasis by downregulating myotubularin-related protein 3 and activating the NF-κB and wnt/β-catenin signaling pathways [39]. The ceramide-induced release of oncogenic exosomal miR-10b from breast cancer cells was indicated to favor tumor progression by suppressing HOXD1 and KLF4 [40]. All of the previously mentioned exosomal microRNAs were shown to be capable of inducing the invasion ability of non-malignant breast cells upon coculture.

Through the process of metastasis, breast cancer cells acquire the ability to transmigrate through blood vessels. One contributor to this process is the tumor-derived exosomal miR-939, which was reported by Di Modica et al. to assist in breast cancer metastasis by regulating cadherin 5 (CDH5) and increasing vascular endothelial cells’ monolayer permeability [41]. A study by Yang et al. showed that miR-146a promoted metastasis by acting on TXNIP and activating the Wnt signaling pathway [42]. Gorczynski et al. reported that antagomirs of exosomal miR-155 and miR-205 from EMT6/4THM breast cancer knock-out mice models (CD200KO and CD200R1KO) impaired tumor growth and metastasis and ameliorated survival in mice, confirming the important roles of these miRNAs in potentiating metastasis [43]. It has been shown that exosomal microRNAs foster the interaction between BC cells and other cells in the tumor niche; mainly fibroblasts. Donnarumma et al. exhibited an important role for CAFs in inducing the stemness and EMT phenotype, promoting metastasis in various breast cancer cell lines by releasing three exosomal microRNAs (miRs-21, -378e, and -143) [28]. Additionally, exosomal miR-9, upregulated in several breast cancer cell lines, was reported to switch the normal fibroblasts (NFs) phenotype into CAFs, thus enhancing tumor growth and metastasis [44]. However, another mode of metastasis is the turbulence of glucose metabolism in the pre-metastatic niche. As demonstrated by Fong et al., miR-122 modifies systemic energy metabolism in favor of disease progression. Mechanistically, miR-122 suppresses glucose uptake by niche cells in vivo and in vitro by downregulating glycolytic enzyme pyruvate kinase and thus increasing nutrient availability in the pre-metastatic niche, enhancing metastasis [45]. In another study on serum exosomal lncRNA small ubiquitin-like modifier 1 pseudogene 3 (SUMO1P3) in (TNBC), a positive correlation was found between serum exosomal SUMO1P3 levels and clinicopathological factors of TBNC patients. Thus, high levels of serum exosomal SUMO1P3 were associated with lymphovascular invasion, lymph node metastasis, and the tumor histological grade [46]. Exosomal microRNAs could also act as inhibitors of tumor progression. Du et al. showed that let-7a inhibited TNBC migration by acting on the c-Myc gene [47]. Park et al. identified three microRNAs—miR-1226-3p, miR-19a-3p, and miR-19b-3p—that inhibit BC migration by regulating aquaporin 5 (AQP5) in MDA-MB231 cells [48]. Moreover, Wei et al. reported that miR-128 inhibits metastasis by downregulating Bax protein in estrogen receptor-positive (ER+) MCF-7 recipient cells [49]. In another study, miR-3613-3p downregulation in fibroblasts led to a marked decrease in the migrating ability of breast cancer cells by binding to the 3′ UTR of SOCS2, a regulator of several signaling pathways, including growth hormone signaling [24].

In addition to the presented general metastasis-promoting abilities of exosomal non-coding RNAs in breast cancer, they were shown to specifically promote distant metastasis to the bone, lung, and brain. Once metastatic breast cancer cells colonize the bone marrow, they hijack signals coming from the normal bone remodeling process and stimulate bone degradation [50]. MiR-19a and integrin-binding sialoprotein (IBSP), significantly expressed and secreted from ER+ BC cells, are active players in this process. An osteoclast-enriched environment is created in the bone by IBSP, stimulating the delivery of exosomal miR-19a that acts on osteoclasts, inducing osteoclastogenesis [51]. Lung metastasis is mediated by exosomal miR-138-5p by promoting the M2 polarization of macrophages through the inhibition of KDM6B [52]. In addition, a high throughput sequencing study on breast-cancer-exosomal lncRNAs revealed that they may be central players in pulmonary pre-metastatic niche formation. Accordingly, BC-derived exosomal lncRNAs were demonstrated to induce lung fibroblast conversion into malignant cells [53]. While the major event of brain metastasis is the destruction of the blood–brain barrier (BBB), brain-metastatic BC-derived exosomes highly expressing the lncRNA GS1-600G8.5 were found to disrupt the BBB via targeting tight junction proteins, permitting the passage of breast cancer cells [54]. In another study on the exosome-mediated lncRNAs effect on the brain metastasis of BC, the loss of the lncRNA XIST was proposed to play a crucial role. A loss of XIST was shown to promote EMT and stemness through a process of protein stabilization [34]. Tumor expansion could also be induced by the interaction between the different types of non-coding RNAs. Yang et al. confirmed that circPSMA1 enhances the tumorigenesis and metastasis of TNBC through the circPSMA1/miR-637/Akt1-β-catenin (cyclin D1) regulatory axis [55]. Zhang et al. have demonstrated that the overexpression of circFOXK2, shown to be upregulated in high metastatic breast cancer cells, could promote migration and invasion by acting with miR-370 and the RNA binding protein IGF2BP3 [35]. 

### 2.3. Roles of Exosomal ncRNAs in Immunoregulation and Cellular Polarization

Exosomal non-coding RNAs have been presented to exhibit distinct immunoregulatory roles in breast cancer, either harboring an oncogenic potential via enabling the immunosuppression/immune evasion of cancer cells or acting as tumor suppressor entities through facilitating an immune response. These are, in turn, achieved primarily via the regulation of the immune checkpoint entity programmed death-ligand 1 (PD-L1) expression, whereby PD-L1 expression enables immunosuppression. Exosomal miR-27a-3p expression was demonstrated to be upregulated in breast cancer following endoplasmic reticulum (ER) stress in macrophages. MiR-27a-3p was presented to be able to confer immune evasion by leading to the upregulation of PD-L1 through relieving the inhibitory effects of MAG12 on PD-L1 via PTEN upregulation, leading to the inactivation of the PI3K/AKT pathway [56]. Similarly, PD-L1 expression was demonstrated to be enhanced via CAFs-derived exosomal miR-92. Mechanistically, immune evasion was achieved via miR-92, which inhibited LATS2, inducing YAP1-mediated PD-L1 transcriptional activation [27]. PD-L1 expression was also presented to be enhanced through the loss of the lncRNA XIST expression in breast cancer, which promoted exosomal miR-503 upregulation, leading to the M1-M2 polarization of microglia and resulting in immunosuppression [34]. In another instance, the exosomal non-coding RNA delivery strategy was proposed as a potential immune-checkpoint blockade tool in TNBC. Exosomal miR-424-5p delivery was shown to suppress PD-L1 signaling, inducing an inflammatory microenvironment and enhancing anti-tumor activity [57]. Additionally, a PD-L1-independent immunosuppression mechanism in TNBC was demonstrated through the miRNA sponging characteristic of exosomal circPSMA1. circPSMA1 was shown to sponge miR-637, leading to the activation of a key immune-related Akt1-β-catenin (cyclin D1) signaling axis, promoting tumorigenesis [55].

In addition to their role in immunoregulation, exosomal non-coding RNAs were documented to contribute to the cellular polarization of macrophages. Macrophages are categorized into M1 and M2, owing to the nature of their cytokine profile and surface markers. M1 macrophages generate an inflammatory response, secreting pro-inflammatory cytokines such as interferon-γ (IFN-γ), tumor necrosis factor-α (TNF-α), interleukin 12 (IL-12), and IL-6. M2 macrophages harbor an immunosuppressive characteristic and induce tumorigenesis, producing anti-inflammatory cytokines, such as transforming growth factor-β (TGF-β) and IL-10 [58]. Macrophages in the tumor microenvironment, i.e., tumor-associated macrophages (TAMs), are the most abundant immune cell type. They adopt distinct M2 phenotypes and harbor a tumor-promoting potential [59]. M1-M2 macrophage cellular polarization was shown to be mediated via the exosomal lncRNA BCRT1, providing them with TAMs-like characteristics and thus promoting the angiogenesis, migration, and immune evasion of BC cells [60]. At the molecular level, the epigenetic factor lysine demethylase 6B (KDM6B) is thought to control macrophage polarization. Exosomal miR-138-5p delivery from BC cells to macrophages was proposed to result in KDM6B downregulation, consequently leading to the M1-M2 polarization of macrophages [52]. Because macrophage M2 polarization was shown to confer a tumor-promoting potential, macrophage repolarization to M1 was proposed as a therapeutic strategy. The treatment of BC cells with epigallocatechin gallate (EECG) was presented to lead to the repolarization of M2 TAMs into M1. EECG-induced miR-16 expression in tumor cells and the exosome-mediated transfer of miR-16 into TAMs resulted in decreased TAM infiltration and M2 polarization, suppressing tumor growth [61]. Macrophage repolarization to M1 was further demonstrated to be mediated via exosomal ncRNAs, namely miR-130 [62] and miR-33 [63], in two studies on 4T1 metastatic breast cancer cells. In both studies, the exosomal delivery of miRNAs (miR-130 and miR-33) was shown to induce M1 repolarization of the macrophages manifested in the upregulation of M1-specific markers and cytokines and the downregulation of M2-specific markers and cytokines. Whereas miR-130 led to a reduced migratory and invasive potential of 4T1 cells along with an improved phagocytotic ability of M1 macrophages, miR-33 resulted in decreased proliferative, invasive, and migratory abilities of 4T1 cells [64]. 

### 2.4. Roles of Exosomal ncRNAs in BC-Drug Sensitivity/Resistance

Recent research on the involvement of exosomal non-coding RNAs in conferring drug sensitivity/resistance has provided evidence on their mechanism of action to affect the response to several types of breast cancer therapeutics, such as trastuzumab-mediated immunotherapy, tamoxifen-mediated hormonal therapy, and several chemotherapeutic agents, such as docetaxel, cisplatin, adriamycin, carboplatin, epirubicin, and gemcitabine. The mode of action of the aforementioned drugs is diverse, where trastuzumab is a monoclonal antibody used to treat human epidermal growth factor receptor 2 (HER2)-positive BCs [65], while tamoxifen is utilized for estrogen receptor (ER)-positive BCs, where it antagonizes estradiol binding to the ER [66]. As for the various chemotherapeutic agents, they generally act to interfere with cellular division, induce DNA damage, and cause cell death [67]. Accordingly, Table 1 summarizes the specific involvement of exosomal non-coding RNAs in conferring BC drug sensitivity/resistance, along with the different mechanisms enabling this phenomenon and the presence of clinical evidence.

In order to confirm that exosomal ncRNAs are the mediators of drug resistance, several studies have adopted the strategy of co-culturing drug-sensitive cells with drug-resistant cells (Figure 2A) or in drug-resistant-cells-derived conditioned media (Figure 2B) in an attempt to demonstrate the exosomal ncRNA-mediated switch from drug sensitivity to resistance [32,68,69,70,71,72,73,74,75,76,77,78,79,80,81,82].

Additionally, another mechanism by which breast cancer cells utilized the exosome-mediated ncRNA transfer in favor of developing drug resistance was the induction of cellular dormancy in the bone marrow microenvironment through mesenchymal stem cells (MSCs) priming. Consequently, dormant cells were able to evade chemotherapeutic agents, leading to metastatic disease recurrence [83,84]. Overall, the exosome-mediated transfer of ncRNAs from resistant to sensitive cells presents a novel therapeutic targeting window for drug-resistant BCs, given the robust evidence on the enhancement of drug resistance. 

**Table 1 ijms-23-08351-t001:** The role of exosomal ncRNAs in drug sensitivity/resistance.

Immunotherapy					
Exosomal ncRNA(s)	Drug	Role	Mechanism	Clinical Evidence and Status	Reference
miR-1246,miR-155	Trastuzumab	Enhancement of trastuzumab resistance	N/A	yes, upregulated	[85]
miR-567	Trastuzumab	Enhancement of trastuzumab sensitivity	Inhibiting autophagy via ATG5 suppression	yes, downregulated	[86]
SNHG14	Trastuzumab	Enhancement of trastuzumab resistance	N/A	yes, upregulated	[77]
AGAP2-AS1	Trastuzumab	Enhancement of trastuzumab resistance	N/A	no	[73]
AFAP1-AS1	Trastuzumab	Enhancement of trastuzumab resistance	Promoting ERBB2 translation via AUF1 binding	yes, upregulated	[72]
**Hormonal Therapy**					
**Exosomal ncRNA(s)**	**Drug**	**Role**	**Mechanism**		**Reference**
miR-221, miR-222	Tamoxifen	Enhancement of tamoxifen resistance	Negatively regulating p27 and ERα	no	[81]
miR-205	Tamoxifen	Enhancement of tamoxifen resistance	Inhibiting apoptosis via E2F1 downregulation	no	[68]
miR-181a-2	Tamoxifen	Enhancement of tamoxifen resistance	Downregulating ERα and activating PI3K/AKT signaling	no	[87]
UCA1	Tamoxifen	Enhancement of tamoxifen resistance	N/A	no	[80]
HOTAIR	Tamoxifen	Enhancement of tamoxifen resistance	N/A	yes, upregulated	[75]
circ_UBE2D2, miR-200a-3p	Tamoxifen	Enhancement of tamoxifen resistance	miR-200a-3p sponging, leading to alterations in cell viability, EMT, and ERα status	no	[71]
**Chemotherapy**					
**Exosomal ncRNA(s)**	**Drug**	**Role**	**Mechanism**	**Clinical Evidence**	**Reference**
multiple miRNAs	Docetaxel	Enhancement of docetaxel resistance	N/A	no	[82]
miR-23b	Docetaxel	Enhancement of docetaxel resistance	Inducing metastatic breast cancer cell dormancy via suppressing MARCKS	yes, upregulated	[84]
miR-134	Cisplatin	Enhancement of cisplatin sensitivity	Negatively regulating STAT5B, Hsp90, and Bcl-2	yes, downregulated	[88]
miR-222	Adriamycin	Enhancement of adriamycin resistance	N/A	no	[79]
miR-222/223	Carboplatin	Enhancement of carboplatin resistance	N/A	no	[83]
miR-1246	Docetaxel, Epirubicin, Gemcitabine	Enhancement of docetaxel, epirubicin, and gemcitabine resistance	Negatively regulating Cyclin-G2	yes, upregulated	[32]
miR-126a	Doxorubicin	Enhancement of doxorubicin resistance	Inducing IL-13^+^ Th2 cells, promoting angiogenesis, and enhancing cell viability via S100A8/A9 upregulation	no	[78]
miR-155	Doxorubicin, Paclitaxel	Enhancement of doxorubicin, and paclitaxel resistance	N/A	no	[76]
miR-423-5p	Cisplatin	Enhancement of cisplatin resistance	N/A	no	[74]
miR-378a-3p, miR-378d	Doxorubicin, Paclitaxel (neoadjuvant)	Enhancement of doxorubicin and paclitaxel resistance	Activation of WNT and NOTCH stemness pathways via DKK3 and NUMB suppression.	yes, upregulated	[69]
HOTAIR	Neoadjuvant chemotherapy	Enhancement of chemoresistance	N/A	yes, upregulated	[75]
H19	Doxorubicin	Enhancement of doxorubicin resistance	N/A	yes, upregulated	[70]

## 3. Diagnostic, Prognostic, and Predictive Biomarker Potential of Exosomal ncRNAs

Given the robust evidence on the role of exosomal ncRNAs in BC onset and progression, they have been presented as novel biomarkers of diagnostic, prognostic, and predictive potential, where they were shown to be readily clinically detectable in several studies. Table 2 outlines the current literature evidence of exosomal ncRNA annotations as diagnostic, prognostic, and predictive biomarkers.

## 4. In Silico Analysis of lnc/circRNA-Sponged miRNAs’ Experimentally Validated Target Genes and Pathways in the BC Exosomal Axis

In an attempt to uncover potential genes and pathways implicated in BC progression and affected by exosomal non-coding RNAs, we took advantage of the already published exosomal ncRNAs dysregulated in BC and the miRNA sponging mechanism of action of lncRNAs and circRNAs and in silico investigated genes and pathways downstream of the BC exosomal lncRNA/circRNA-miRNA-target axis. Accordingly, we utilized the miRTargetLink 2.0 [104] tool to uncover potential shared genes and pathways that are experimentally validated to be targeted by miRNAs, sponged by either lncRNAs or circRNAs in the BC exosomal axis, where these ncRNAs were selected from the reviewed literature. MirTargetLink incorporates experimental validation (strong/weak) information for genes from miRTarBase 8.0 [105]. Strong and weak validations are attributed to the respective assays, where luciferase reporter assay, Western blot, and qPCR are considered as strong evidence whereas microarray, NGS, pSILAC, CLIP-seq, and others as weak evidence. As for the pathways information, miRTargetLink incorporates annotations from mirPathDB 2.0 [106] for strong and weak experimentally validated miRNA pathways. After manually curating the literature, miRNAs (hsa-miR-503 [34], hsa-miR-330-5p [25], hsa-miR-16-5p [102], hsa-miR-106a-5p [107], hsa-miR-370 [35], hsa-miR-637 [55], and hsa-miR-200a-3p [71]) were obtained and were sponged by their respective lncRNAs/circRNAs (XIST, SNHG3, SNHG16, HAND2-AS1, circFOXK2, circPSMA1, and circUBE2D2) in the BC exosomal axis (Figure 3). This miRNA set was used as an input for the miRTargetLink tool, where the authors sought after genes that are strongly experimentally validated to be targeted by a minimum of two miRNAs from our miRNA set, ending up with a set of 20 strong validated genes (Figure 4). Interestingly, the vast observed literature indicated that all of the resulted genes were implicated in BC pathology at some point [108,109,110,111,112,113,114,115,116,117,118,119,120,121,122,123,124,125,126,127].

In addition, in order to identify shared pathways among our miRNA set, a similar approach was followed, though with increasing the minimum targeting threshold from two to four miRNAs to narrow down our results, along with obtaining weak and strong experimental evidence since the tool would not permit choosing only strong or weak. A set of 117 shared pathways were obtained, containing BC, integrated BC, and ErbB signaling pathways (Figure 5), and this is supportive of the implication of the exosomal non-coding RNA axis in BC pathology.

In an attempt to further explore the functional significance of the resultant strong validated gene set and investigate their functional implication in BC and other pathways, the DAVID functional annotation and enrichment analysis tool (2021 update) [128] was used to enable the authors to conduct gene ontology [129] (GO) terms enrichment analysis and Kyoto Encyclopedia of Genes and Genomes [130] (KEGG) pathway analysis, among others. For both analyses, the authors set a minimum gene count threshold of 5 out of the total 20 resultant strong validated genes, along with an FDR cut-off of 0.05 in order to narrow down the list of significant GO terms and KEGG pathways.

GO terms enrichment analysis classifies the results into three subsets of GO annotations, namely biological process (BP), molecular function (MF), and cellular component (CC). Table 3 outlines the significant GO terms associated with our gene set for each GO term annotation subset.

The KEGG pathway analysis of our gene set resulted in a list of significant KEGG pathways that are represented in Table 4. Interestingly, the BC pathway was found to be one of the significant KEGG pathways, where 8 of the 20 total genes were found to be implicated in BC pathogenesis at different levels, subtypes, and signaling pathways (Figure 6). In addition, among the notable significant KEGG pathways related to BC were the PI3K-Akt signaling pathway, p53 signaling pathway, cell cycle, and MAPK signaling pathway. Moreover, supporting the well-documented role of exosomal non-coding RNAs in BC drug resistance, EGFR tyrosine kinase inhibitor resistance and endocrine resistance were also significant KEGG pathways associated with our selected gene set.

## 5. Search Strategy 

The search strategy was separated into three approaches: targeting miRNAs, lncRNAs, and circRNAs in BC exosomes using the PubMed medical subject heading (MeSH) database. For exosomal miRNAs in BC, PubMed searched for the following: “Breast Neoplasms” [MeSH] AND “Exosomes” [MeSH] AND “MicroRNA” [MeSH]. For exosomal lncRNAs in BC: “Breast Neoplasms” [MeSH] AND “Exosomes” [MeSH] AND “RNA, long non-coding” [MeSH]. As for exosomal circRNAs in BC: “Breast Neoplasms” [MeSH] AND “Exosomes” [MeSH] AND “Circular RNA” [MeSH]. All of the MeSH terms found below the abovementioned MeSH terms were also included. Other databases, such as MEDLINE Ovid and Embase, were also searched using the same strategy, where the same set of articles was returned. Therefore, we concentrated our review on the results of PubMed searches. In addition, information on the number of publications, along with signature publications and the discovery timeline of exosomal ncRNAs in BC, are presented in (Figure 7).

## 6. Challenges

Exosome research has witnessed a rapid growth and exosomes have shown to be a great platform in diagnostic, prognostic, and therapeutic applications. However, there are certain challenges that hinder the progress in this field and should be targeted. One major challenge in exosomal research is the absence of a competent standardized isolation and purification method. Despite several techniques being available, ultracentrifugation being the most common one, there are limitations that affect the yield, diversity, purity, and function of the isolated exosomal product [131,132,133]. To begin with, ultracentrifugation, which is the gold standard method that isolates exosomes according to their density, is time-consuming, cumbersome, and yields a product with a low purity and different morphology [131,132,134,135]. Flow-cytometry-based analyses suggest that the isolated exosome fractions from this technique are often contaminated by co-isolated plasma proteins. The latter may haggle the precision of exosome-based diagnosis [136]. Other techniques, such as size exclusion chromatography, ultrafiltration, and immune-affinity, that separate exosomes according to size and function are exorbitant and impotent in removing impurities [135]. Another challenge is exosome quantification [135]. Current techniques include, among others, nanoparticle tracking analysis, flow cytometry, tunable resistive pulse sensing, electron microscopy, dynamic light scattering, microfluidics-based detection, and surface plasmon resonance. However, all of these techniques yield inaccurate results because none of them have the required high sensitivity to detect exosomes of all sizes, except for electron microscopy, where exosomes are counted manually. However, this needs elbow grease and is time consuming. In addition, some of the exosomes may be lost during sample preparation [134]. To elaborate more on the issue of non-sensitivity, flow cytometry machines could miss exosomes below 300 nm in size due to their side detection limitation [134]. Apart from the low sensitivity, nanoparticle tracking analysis, flow cytometry, and surface plasmon resonance need the use of high-cost equipment [134].

The study of non-coding RNAs has shown several challenges as well. First, regarding research on microRNAs as biomarkers and knowing that each subtype of BC has its specific array of microRNA, the selected patients in a study should be the ones having the least variability in the subtype of BC, histotype, ethnicity, and age at diagnosis, due to these being the major factors affecting the microRNA profile of each patient. Another challenge in this field is in the experimental design, where many studies have a poor one. The third challenge is in the isolation of miRNA. miRNAs are usually isolated as total RNA using guanidine/phenol/chloroform-based protocols, where the RNA yield in serum and plasma is usually low. Moreover, there are limitations in the detection of microRNAs. Different techniques are used for the detection of miRNAs. Utilizing microarray and in situ hybridization is not enough to assign miRNA as biomarkers for early breast cancer (EBC) since these platforms have a low sensitivity. In addition, sequencing is still an expensive assay to be routinely used for miRNA detection. The most commonly used method is a RT-qPCR assay for the specific validation combined with a statistical test that defines the accuracy of the results. However, the absence of a standard endogenous control is still a major limitation in this assay. Finally, it is worth stating that the selection of one microRNA as a biomarker is not adequate but that a panel of microRNAs would be more definitive [137]. 

Despite the promising results, research on exosomal LncRNAs is still in its infancy, with several limitations, as more studies must be conducted to unveil and elaborate their specific relation to diseases, pathways, and their underlying molecular mechanisms [133].

As for circRNA, more unprejudiced techniques, such as RNA sequencing, should be used to discover novel circRNA. However, this technique has certain limitations that hinder its accuracy. To elaborate more, the results are highly dependent on the protocol chosen for library preparation and on RNA quality. Secondly, RT-qPCR is a potent method for authenticating genome-wide data. However, an appropriate normalization of RT-qPCR data is challenging, and the use of a single, unvalidated reference gene may lead to an unreliable conclusion. Finally, future studies should explore more about the novel technologies for circRNA detection, such as nanopore RNA sequencing, which can provide information on the entire circRNA [138].

## 7. Conclusions and Recommendations

In this paper, an experimentally validated BC exosomal non-coding RNAs-regulated target gene axis was established, where 20 signature genes that are downstream of this axis were pinpointed, and literature evidence demonstrated the involvement of these genes at some point during BC pathology. In addition, this target gene axis was shown to be implicated in BC-related pathways, supporting our hypothesis. Furthermore, GO and KEGG pathway analyses on the 20 signature genes showed promising associations with different BC subtypes and various BC-related signaling and drug resistance pathways. Hence, we propose that this target gene set could be a starting point for future studies, aiming to better understand and interpret the different stages of BC pathology and mechanisms of drug resistance. 

As future perspectives, more research should be carried out to target the challenges in exosome research and establish a standardized protocol for the exosomal non-coding RNA assessment from sample collection to data analysis. Regarding exosome isolation, there is a need to find an ascendible, reproducible, time-efficient, and cost-effective isolation protocol [136]. Moreover, limitations in exosome quantification may be solved by updating the flow cytometers to have multi-angle lasers for a better resolution of exosomes [134]. In addition, future studies should explore more about novel technologies and techniques for standard and proper detection and isolation methods for exosomal non-coding RNAs [137,138]. Finally, future experimental validations should be conducted to address the clinical value of the experimentally validated breast cancer exosomal non-coding RNAs-regulated target gene axis that we identified in BC. Specifically, further research will be needed to validate this gene set and the specific ncRNAs implicated in the related axis and their potential in the development of novel diagnostic, prognostic, or therapeutic strategies against BC.

## Figures and Tables

**Figure 1 ijms-23-08351-f001:**
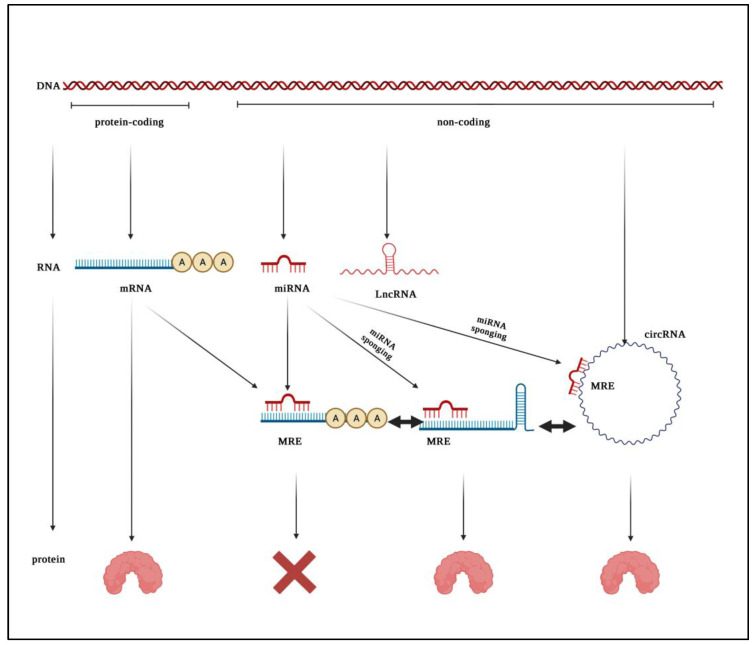
MiRNA sponging by circRNAs and lncRNAs. The central dogma proposes that protein-coding regions of the genome give rise to mRNAs, which are, in turn, translated into proteins. However, miRNAs, which are transcribed from the non-coding genome, could bind mRNAs via MREs and inhibit their translation. LncRNAs and, circRNAs, which are also transcribed from the non-coding genome, could bind and sponge miRNAs since they harbor MREs, resulting in the translation of the previously miRNA-inhibited mRNAs into proteins. MRE: MicroRNA response element. Figure was created with BioRender.com (accessed date: 24 July 2022).

**Figure 2 ijms-23-08351-f002:**

The exosomal ncRNAs-mediated switch of BC cells from drug sensitivity to resistance. (**A**) Co-culture of drug-resistant with drug-sensitive cells leads to an exosomal ncRNAs-mediated switch into drug resistance. (**B**) Culture of drug-sensitive cells in drug-resistant BC-cells-conditioned media, which contains drug-resistant cell-secreted exosomes, leads to an exosomal ncRNAs-mediated switch into drug resistance.

**Figure 3 ijms-23-08351-f003:**
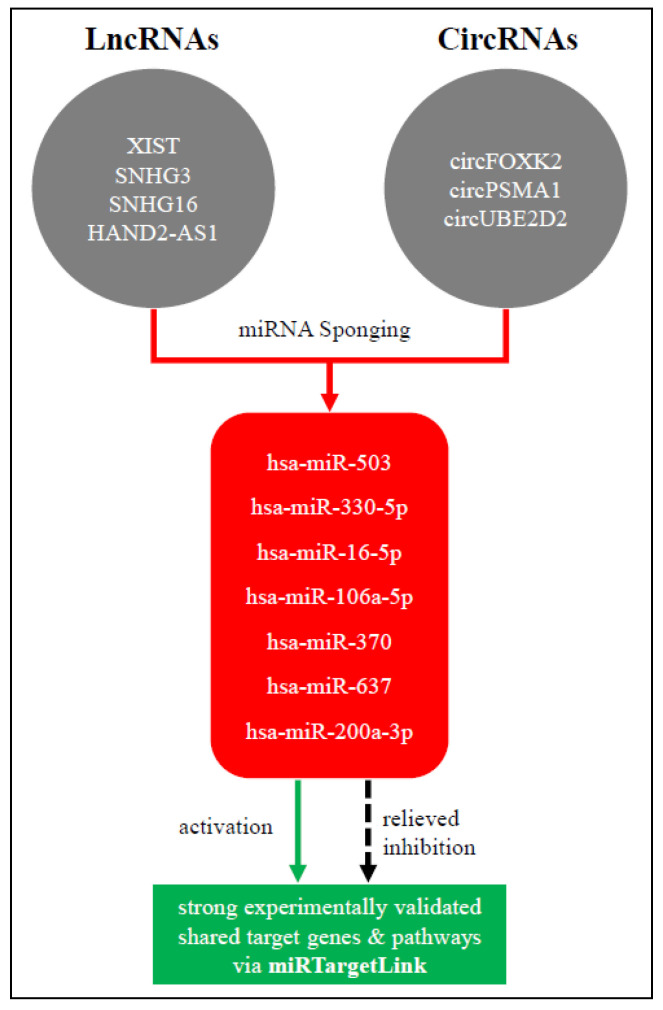
The breast cancer exosomal lncRNA/circRNA-miRNA-target axis.

**Figure 4 ijms-23-08351-f004:**
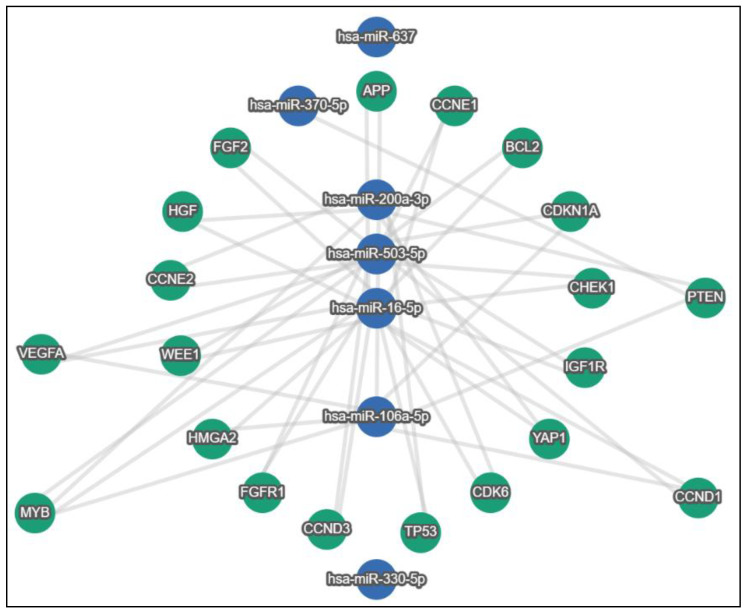
Strongly experimentally validated shared target genes downstream of the BC exosomal lncRNA/circRNA-miRNA-target axis. Green nodes indicate genes and blue nodes indicate miRNAs. Minimum miRNA threshold = 2.

**Figure 5 ijms-23-08351-f005:**
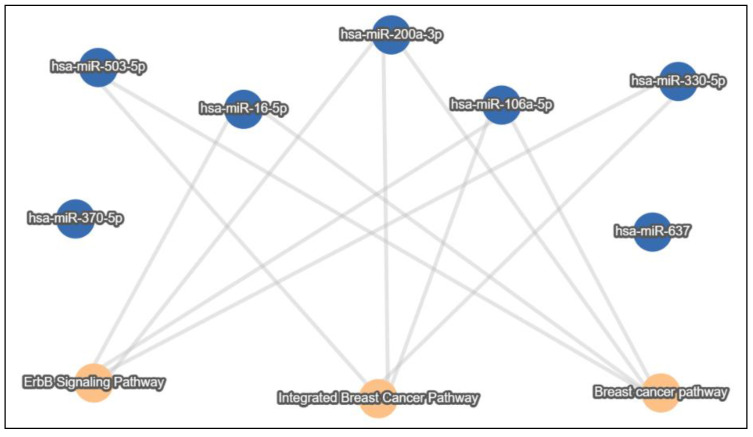
Strongly and weakly experimentally validated BC-related shared target pathways downstream of the BC exosomal lncRNA/circRNA-miRNA-target axis. Orange nodes indicate pathways and blue nodes indicate miRNAs. Minimum miRNA threshold = 4. Only BC-related pathways are shown.

**Figure 6 ijms-23-08351-f006:**
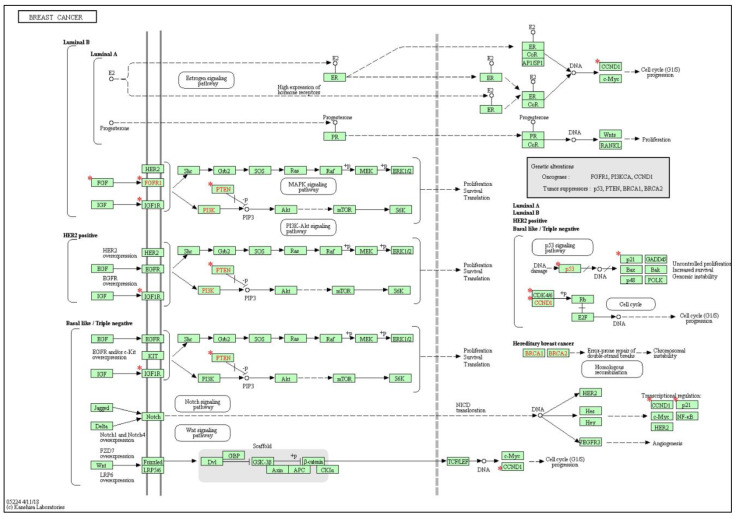
Implication of the strong experimentally validated shared target genes downstream of the BC exosomal lncRNA/circRNA-miRNA-target axis in BC KEGG pathway. Eight out of the twenty shared target genes were found to be implicated in BC KEGG pathway. Red stars indicate the shared target genes. Red font color indicates signature genetic alterations.

**Figure 7 ijms-23-08351-f007:**
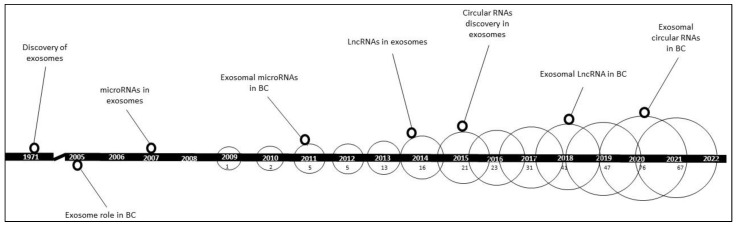
A timeline figure showing the discoveries and increase in interest reflected by the number of publications about exosomal non-coding RNAs in BC. # in the circles denotes the number of publications in PubMed per year.

**Table 2 ijms-23-08351-t002:** Diagnostic, prognostic, and predictive biomarker potential of exosomal ncRNAs in BC.

Noncoding RNA	Source	Biomarker Type	Status	Clinical Evidence	Mechanism	Reference
miR-421, miR128-1, miR128-2	plasma	Diagnostic/Prognostic	Upregulated	Yes	N/A	[89]
miR-3662, miR-146a, miR-1290	serum	Predictive/Diagnostic	Upregulated	Yes	N/A	[90]
miR-424, miR-423, miR-660, let7-i	urine	Diagnostic	Up, down, down, downregulated	Yes	N/A	[91]
miR-148a	serum	Diagnostic/Prognostic	Downregulated	Yes	N/A	[92]
miR-17-5P	serum	Diagnostic	Downregulated	Yes	N/A	[93]
miR-1246, miR-155	serum	Predictive/Prognostic	Upregulated	Yes	N/A	[85]
miR-1910-3p	serum	Diagnostic	Upregulated	Yes	downregulated myotubularin-related protein 3, activated the NF-κB and wnt/β-catenin signaling pathway, and promoted breast cancer progression	[39]
miR-1976	plasma, tissues	Diagnostic	Downregulated	Yes	N/A	[94]
miR-21 (with MMP1)	urine	Diagnostic	Downregulated, Upregulated	Yes		[95]
Let-7b-5p, miR-122-5p, miR-146b-5p, miR-210-3p, miR-215-5p	plasma	Diagnostic	N/A	No	N/A	[96]
miR-21, miR-222, miR-155	serum	Diagnostic/Predictive	Upregulated	Yes	N/A	[97]
miR-16, miR-30b, miR-93	serum, plasma	Diagnostic	Up, down, upregulated	Yes	N/A	[98]
miR-106a-3p, miR-106a-5p, miR-20b-5p, miR-92a-2-5p	plasma, serum	Diagnostic	Upregulated	Yes	N/A	[99]
miR-1246	serum	Diagnostic	Upregulated	No	Suppresses the expression of cyclin-G2 (CCNG2)	[32]
miR-1246, miR-21	plasma	Diagnostic	Upregulated	Yes	N/A	[100]
SNHG14	serum	Diagnostic	Upregulated	Yes	N/A	[77]
HOTAIR	serum	Diagnostic/Prognostic	Upregulated	Yes	N/A	[75]
HOTAIR	plasma	Diagnostic/Prognostic	Upregulated	Yes	Positively correlated with ERBB2/HER2 expression	[101]
AFAP1-AS1	serum	Diagnostic/Prognostic	Upregulated	Yes	Promotes ERBB2 translation via AUF1 binding	[72]
SNHG16	peripheral blood	Prognostic	Upregulated	Yes	Promotes CD73 expression on γδ1 T cells via the TGF-β1/SMAD5 pathway, enabled via miR-16-5p sponging	[102]
H19	serum	Diagnostic/Prognostic	Upregulated	Yes	N/A	[70]
SUMO1P3	serum	Diagnostic/Prognostic	Upregulated	Yes	N/A	[46]
circFOXK2	tissues	Diagnostic	Upregulated	No	Acts with IGF2BP3 and miR370	[35]
circPSMA1	serum	Prognostic	Upregulated	Yes	circPSMA1 sponges miR-637, activating Akt1-β-catenin (Cyclin D1) signaling	[55]
hsa-circRNA-0005795, hsa-circRNA-0088088	serum	Diagnostic	Downregulated, Upregulated	Yes	N/A	[103]

**Table 3 ijms-23-08351-t003:** Significant gene ontology (GO) terms associated with strong experimentally validated shared target genes downstream of the BC exosomal lncRNA/circRNA-miRNA-target axis. Minimum gene count: 5. FDR cut-off: 0.05.

Biological Process		
GO Term	Gene Count	FDR
negative regulation of transcription from RNA polymerase II promoter	10	4.70 × 10^−5^
regulation of cell cycle	7	4.70 × 10^−5^
regulation of cyclin-dependent protein serine/threonine kinase activity	5	4.70 × 10^−5^
positive regulation of gene expression	8	4.70 × 10^−5^
negative regulation of G1/S transition of mitotic cell cycle	5	4.70 × 10^−5^
positive regulation of MAPK cascade	6	4.70 × 10^−5^
negative regulation of apoptotic process	8	4.70 × 10^−5^
cytokine-mediated signaling pathway	7	4.70 × 10^−5^
G1/S transition of mitotic cell cycle	5	6.50 × 10^−5^
positive regulation of protein kinase B signaling	6	8.00 × 10^−5^
cell division	7	8.80 × 10^−5^
positive regulation of protein phosphorylation	6	8.80 × 10^−5^
positive regulation of phosphatidylinositol 3-kinase signaling	5	1.10 × 10^−4^
cellular response to DNA damage stimulus	6	2.40 × 10^−4^
negative regulation of cell proliferation	6	3.00 × 10^−3^
positive regulation of transcription from RNA polymerase II promoter	8	4.00 × 10^−3^
protein phosphorylation	6	4.30 × 10^−3^
positive regulation of cell proliferation	6	5.20 × 10^−3^
response to drug	5	5.40 × 10^−3^
negative regulation of gene expression	5	7.50 × 10^−3^
nervous system development	5	1.40 × 10^−2^
positive regulation of transcription, DNA-templated	5	6.30 × 10^−2^
Molecular Function		
GO Name	Gene Count	FDR
protein kinase binding	7	2.90 × 10^−4^
identical protein binding	9	4.40 × 10^−3^
protein binding	20	1.20 × 10^−2^
protein serine/threonine/tyrosine kinase activity	5	1.50 × 10^−2^
Cellular Component		
GO Name	Gene Count	FDR
cyclin-dependent protein kinase holoenzyme complex	6	2.30 × 10^−8^
nucleus	18	3.20 × 10^−6^
nucleoplasm	14	1.20 × 10^−4^
cytoplasm	15	6.10 × 10^−4^
centrosome	5	2.30 × 10^−2^
macromolecular complex	5	3.90 × 10^−2^
membrane	8	5.90 × 10^−2^
extracellular region	6	3.20 × 10^−1^

**Table 4 ijms-23-08351-t004:** Significant KEGG pathways associated with strongly experimentally validated shared target genes downstream of the BC exosomal lncRNA/circRNA-miRNA-target axis. Minimum gene count: 5. FDR cut-off: 0.05. Unrelated KEGG pathways were eliminated.

KEGG Pathway	Gene Count	FDR
PI3K-Akt signaling pathway	15	4.40 × 10^−14^
p53 signaling pathway	10	6.60 × 10^−13^
Pathways in cancer	14	1.10 × 10^−10^
Cell cycle	9	2.00 × 10^−9^
Cellular senescence	9	8.10 × 10^−9^
MicroRNAs in cancer	10	7.50 × 10^−8^
Breast cancer	8	1.50 × 10^−7^
Proteoglycans in cancer	8	1.10 × 10^−6^
EGFR tyrosine kinase inhibitor resistance	6	2.80 × 10^−6^
Focal adhesion	7	1.40 × 10^−5^
Endocrine resistance	5	1.80 × 10^−4^
MAPK signaling pathway	6	1.00 × 10^−3^
Rap1 signaling pathway	5	2.50 × 10^−3^
Chemical carcinogenesis-receptor activation	5	2.50 × 10^−3^
Ras signaling pathway	5	3.30 × 10^−3^

## Data Availability

Not applicable.

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
