# Peer review of "Spotlight on Exosomal Non-Coding RNAs in Breast Cancer: An In Silico Analysis to Identify Potential lncRNA/circRNA-miRNA-Target Axis"

_ijms, 2022, doi:10.3390/ijms23158351_

Round 1

Reviewer 1 Report

This review summarizes the state-of-the-art knowledge regarding noncoding RNAs in breast cancer and also performs bioinformatic analysis of the findings.

It is well written, and the outcome is a very comprehensive review.

I have only few points that are suggested to be corrected:

1.       The biogenesis of exosomes is irrelevant, please delete lines 61-68.

2.       Please spell out: HMLE cells (line 145)

3.       In lines 163-172: the source of the mentioned circs should be depicted.

4.       Please spell out TNBC- line 271.

5.       Table 1 and Table 2 are summarized in chapters 2.4 and 3, respectively. To avoid redundancy, I would either shorten these chapters or move the tables to the supplementary material.

6.       Lines 459-461 should be rephrased

7.       Figure 2 is somewhat misleading, with regards to the outcome of sponging miRs by Lncs and circs, maybe add the word sponging in the two diagonal arrows or add numbers to these arrows and explain in the legend.

Author Response

Thank you for these comments. We have responded and modified requested changes below

  1. The biogenesis of exosomes is irrelevant, please delete lines 61-68.

This paragraph was deleted as per the reviewer’s suggestion.

  1. Please spell out: HMLE cells (line 145)

We have now included in the revised manuscript the full name of the cell line: Human Mammary Epithelial Cells.                                         

  1. In lines 163-172: the source of the mentioned circs should be depicted.

Thank you for pointing this out. The source of the mentioned circs i.e., their exosomal origin was added.

  1. Please spell out TNBC- line 271.

       We have already spelled out TNBC as Triple Negative Breast Cancer at its first appearance in the text in line 144.  

  1. Table 1 and Table 2 are summarized in chapters 2.4 and 3, respectively. To avoid redundancy, I would either shorten these chapters or move the tables to the supplementary material.

      As per the suggestion of the reviewer, we have now removed the summaries to avoid redundancy.  

  1. Lines 459-461 should be rephrased

 Thank you for pointing this out. We have now rephrased the sentence to make it clearer.

The new sentence is “The search strategy was separated into three approaches , targeting miRNAs, lncRNAs and circRNAs in BC exosomes using the PubMed medical subject heading (MeSH) database”.

  1. Figure 2 is somewhat misleading, with regards to the outcome of sponging miRs by Lncs and circs, maybe add the word sponging in the two diagonal arrows or add numbers to these arrows  and explain in the legend.

Figure 2 is now revised. We added the “miRNA Sponging” phrase to the two diagonal arrows as per your suggestion.

Reviewer 2 Report

Thank you for submitting the article to the journal.  

It is a very comprehensive and up-to-date article. I have a few suggestions that would further enhance the article's credibility:  

1. In discussion, please present a tabular presentation of the available literature.

2. How can it be to clinical use and any trials currently going on should be added in the article. It can be assessed on the NIH website.  

3. Please separate the challenges and recommendation section. Instead, I suggest reviewing available literature and presenting it as clinically relevant as possible, which could direct future researchers.  

4. Add a section in the discussion labeled as future perspective and add how can the content be useful in another clinical scenario besides breast cancer.  

5. figure 7 is impressive. please zoom it in and take care that the small fonts are not distorted.

Author Response

Thank you for the time and efforts taken in reviewing our manuscript.

  1. In discussion, please present a tabular presentation of the available literature.

 We have tabulated the available literature extensively in tables 1 and 2 and also within the manuscript as in-text citations. However, to address your comment, we have referred to these tables in the discussion, when applicable.

  1. How can it be to clinical use and any trials currently going on should be added in the article. It can be assessed on the NIH website.  

Thank you for this comment. We have searched the NIH website here: https://clinicaltrials.gov/ as per your suggestion. However, the field of exosomal non-coding RNA is new and there does not seem to be any clinical trials currently for “exosomal noncoding RNAs in BC”. We have tried different search strategies too, for example, “non-coding RNAs in BC” has returned 4 hits, but none on  exosomal non-coding RNAs strictly.

We believe that the in-silico analysis that we performed on exosomal non-coding RNAs in Breast Cancer and the review that we present throughout our article will definitely guide future studies that will address their clinical value as we point out to their involvement in drug sensitivity/resistance and as diagnostic, prognostic and predictive biomarkers in tables 1 and 2, respectively. This is a topic that will be gain more interest and clinical application in the future. However, further experimental validations and research should be conducted before they can reach to the point of clinical trials.

  1. Please separate the challenges and recommendation section. Instead, I suggest reviewing available literature and presenting it as clinically relevant as possible, which could direct future researchers.  
  2. Add a section in the discussion labeled as future perspective and add how can the content be useful in another clinical scenario besides breast cancer.

We addressed both suggestions (3, and 4) collectively as we separated the challenges from the recommendations part and added the recommendations to the conclusion, where we have also included the future perspectives of the clinical implications of exosomal non-coding RNAs in BC, other cancers and clinical scenarios to direct future researchers’ work.

  1. Figure 7 is impressive. please zoom it in and take care that the small fonts are not distorted.

Thank you! We optimized the figure quality accordingly such that it is perfectly zoomable.